# eHBB: a randomised controlled trial of virtual reality or video for neonatal resuscitation refresher training in healthcare workers in resource-scarce settings

Rachel Umoren ,[1] Sherri Bucher,[2] Daniel S Hippe,[3] Beatrice Nkolika Ezenwa,[4] Iretiola Bamikeolu Fajolu,[4] Felicitas M Okwako,[5] John Feltner,[1] Mary Nafula,[5] Annet Musale,[5] Olubukola A Olawuyi,[4] Christianah O Adeboboye,[4] Ime Asangansi,[6] Chris Paton,[7] Saptarshi Purkayastha,[8] Chinyere Veronica Ezeaka,[4] Fabian Esamai[5]

RU and SB contributed equally. CVE and FE contributed equally.

RU and SB are joint first authors. CVE and FE are joint senior authors.

For numbered affiliations see end of article.

**Correspondence to**
Dr Rachel Umoren;
rumoren@uw.edu

## ABSTRACT

**Objective** To assess the impact of mobile virtual reality (VR) simulations using electronic Helping Babies Breathe (eHBB) or video for the maintenance of neonatal resuscitation skills in healthcare workers in resource-scarce settings.

**Design** Randomised controlled trial with 6-month follow-up (2018–2020).

**Setting** Secondary and tertiary healthcare facilities.

**Participants** 274 nurses and midwives assigned to labour and delivery, operating room and newborn care units were recruited from 20 healthcare facilities in Nigeria and Kenya and randomised to one of three groups: VR (eHBB +digital guide), video (video +digital guide) or control (digital guide only) groups before an in-person HBB course.

**Intervention(s)** eHBB VR simulation or neonatal resuscitation video.

**Main outcome(s)** Healthcare worker neonatal resuscitation skills using standardised checklists in a simulated setting at 1 month, 3 months and 6 months.

**Results** Neonatal resuscitation skills pass rates were similar among the groups at 6-month follow-up for bag-and-mask ventilation (BMV) skills check (VR 28%, video 25%, control 22%, p=0.71), objective structured clinical examination (OSCE) A (VR 76%, video 76%, control 72%, p=0.78) and OSCE B (VR 62%, video 60%, control 49%, p=0.18). Relative to the immediate postcourse assessments, there was greater retention of BMV skills at 6 months in the VR group (−15% VR, p=0.10; −21% video, p<0.01, −27% control, p=0.001). OSCE B pass rates in the VR group were numerically higher at 3 months (+4%, p=0.64) and 6 months (+3%, p=0.74) and lower in the video (−21% at 3 months, p<0.001; −14% at 6 months, p=0.066) and control groups (−7% at 3 months, p=0.43; −14% at 6 months, p=0.10). On follow-up survey, 95% (n=65) of respondents in the VR group and 98% (n=82) in the video group would use their assigned intervention again.

**Conclusion** eHBB VR training was highly acceptable to healthcare workers in low-income to middle-income countries and may provide additional support for neonatal

## Strengths and limitations of this study

► This study was a multicentre, randomised controlled trial of mobile virtual reality or video to support neonatal resuscitation skills retention in nurses and midwives who provide neonatal resuscitation in two low-income to middle-income countries.

► The study used an evidence-based Helping Babies Breathe second edition curriculum designed for neonatal resuscitation training in resource-scarce settings.

► Healthcare workers accessed the digital interventions on mobile phones in the 6 months following in-person Helping Babies Breathe training.

► Healthcare workers were recruited from secondary and tertiary healthcare settings in urban and semi-urban resource-scarce settings, so the study findings may not apply to healthcare workers in high-resource, primary healthcare or rural settings.

resuscitation skills retention compared with other digital interventions.

## INTRODUCTION

In 2019, there were 2.4 million deaths among infants under 28 days of age.[1 2] These neonatal deaths now account for 47% of global under 5 years child mortality,[2] and most cases are preventable.[3–5] The majority of these deaths occur in low-income to middle-income countries (LMICs). Nigeria, with a neonatal mortality rate of 36 deaths per 1000 live births and Kenya at 21 deaths per 1000 live births in 2019,[1] are at significant risk of failing to meet the United Nations Sustainable Development Goal 3 to reduce neonatal mortality to 12 per 1000 live births by 2030.[3]

Intrapartum asphyxia or lack of breathing at birth is a leading cause of neonatal mortality.[6 7] Training healthcare workers in neonatal resuscitation using the *Helping Babies Breathe* (HBB) curriculum builds competency and reduces newborn morbidity and mortality.[8–13] However, neonatal resuscitation skills are quickly lost after trainings that use the traditional approach of small group facilitated classroom training.[14–19] For this reason, periodic refresher training with 'low-dose high frequency' manikin-based simulations are recommended to support neonatal resuscitation skills retention.[20 21] Unfortunately, access to manikin-based simulation is limited by trainer, space and equipment availability.[22 23] Yet, the high penetration of smartphones and cellular network connectivity in urban and rural areas in LMICs, makes innovative simulation training feasible using mobile virtual reality (VR) simulations for healthcare workers (HCWs) who provide care in health facility and community-based settings.

VR simulations are effective educational tools and can be engaged at the learner's convenience, on their own smartphone, with game-based automated feedback that is ideal for episodic learning.[24–26] However, little is known of their feasibility, acceptability or effectiveness for neonatal resuscitation skills retention in LMICs. We hypothesised that mobile VR simulation refresher training would address challenges to the quality of newborn resuscitation related to the maintenance of HCW knowledge and skills over time addressing the lack of standardised dissemination of updates to recommended practice and high rates of staff turnover. The objective of this study was to evaluate the impact of eHBB VR used with in-person neonatal resuscitation training on neonatal resuscitation educational indicators and performance outcomes, in comparison to other digital refresher training modalities.

## METHODS
### Study setting
The study was conducted in Lagos, Nigeria and Busia, Western Kenya. Twelve healthcare facilities (nine secondary and three tertiary) were located in Nigeria while eight facilities were located in Kenya. The healthcare facilities were located in urban and semi-urban areas and all have maternal and newborn services with newborn bed capacity ranging from 2 to 80 beds and delivery and neonatal unit staffing capacity from 7 to 124 nurses (see online supplemental file 1).

### Participants
Study participants consisted of nurses and nurse-midwives assigned to labour and delivery, operating room and newborn care units. Site coordinators or research assistants requested contact numbers, units and wards of potential participants from head nurses at identified facilities. Research assistants contacted individuals to determine eligibility and obtained consent (see online supplemental file 2).

### Inclusion criteria
Nurses and midwives who participate in deliveries and who provide neonatal resuscitation to inborn or outborn infants and provide study consent.

### Exclusion criteria
Those who had attended a neonatal resuscitation training course in the 1 year preceding the study; individuals who did not provide neonatal resuscitation as part of their duties or would be unavailable or unwilling to participate in follow-up study activities throughout the 6-month postinitial training period.

### Randomisation
Study IDs generated for each country site were randomly assigned via a computer-generated algorithm to the VR, video and control groups by a US-based study coordinator. Participants were enrolled and assigned a study ID before the HBB course by local study coordinators. Each participant received an Android study phone, preloaded with permission-based access linked to their study ID, via the mobile Helping Babies Survive powered by District Health Information System (DHIS2) app (mHBS/DHIS2), to the participant's assigned digital intervention. The data analysis team was blinded to the study assignments.

### HBB course structure
The HBB provider course (second edition)[27] was taught by study HBB master trainers as 1 day, 8-hour long sessions from December 2018 to August 2019. A 30 min orientation was provided on use of the mHBS/DHIS2 app, including how to access the assigned digital intervention. All participants had access to a digitised HBB provider manual through the mHBS/DHIS2 app. The VR group in addition accessed the eHBB VR simulations which consisted of three interactive three-dimensional simulation scenarios representing care of a newborn requiring routine care, some resuscitation and prolonged resuscitation with positive pressure ventilation. The features of eHBB VR have been previously described and the application is available for free download.[26 28] The neonatal resuscitation video used by the video group featured preparation for delivery and the resuscitation of a newborn requiring positive pressure ventilation.[29] None of the interventions required internet for use. A total of 274 HCWs participated in the in-person HBB training.

### Precourse and postcourse assessments
Standardised knowledge and skills assessments were conducted by trained research assistants. The HBB knowledge check (15 of 18 multiple-choice questions, ≥80% required to pass) and bag-and-mask ventilation skill check (BMV; 14 of 14 items required to pass) were conducted precourse and postcourse along with the objective structured clinical examination (OSCE) A checklist on preparation for delivery and initial steps of resuscitation (9 out of 12 items and 3 required items to pass). In addition, the postcourse assessment included the OSCE B checklist on prolonged newborn resuscitation (17 out of 23 items and

| Study group | Precourse period | | | Follow-up period | | | |
|---|---|---|---|---|---|---|---|
| | Baseline assessment | Randomization to study intervention | Precourse assessment | Immediate postcourse assessment | 1-month postcourse assessment | 3-month postcourse assessment | 6-month postcourse assessment |
| VR | Knowledge test | VR + Digital HBB provider's guide | Knowledge test, simulation performance | Knowledge test, simulation performance | Knowledge test, simulation performance | Knowledge test, simulation performance | Knowledge test, simulation performance |
| | | | | | VR + Digital HBB provider's guide + LDHF manikin practice | | |
| Video | Knowledge test | Video + Digital HBB provider's guide | Knowledge test, simulation performance | Knowledge test, simulation performance | Knowledge test, simulation performance | Knowledge test, simulation performance | Knowledge test, simulation performance |
| | | | | | Video + Digital HBB provider's guide + LDHF manikin practice | | |
| Standard practice (control) | Knowledge test | Digital HBB provider's guide only | Knowledge test, simulation performance | Knowledge test, simulation performance | Knowledge test, simulation performance | Knowledge test, simulation performance | Knowledge test, simulation performance |
| | | | | | Digital HBB provider's guide + LDHF manikin practice | | |

**Figure 1** Study diagram. BMV, bag-and-mask ventilation; HBB, helping babies breathe; LDHF, low-dose high frequency; VR, virtual reality.

6 required items to pass). HBB checklists are available for free download from the American Academy of Pediatrics.[30] A demographic survey was completed (figure 1).

### Postcourse interventions and follow-up

Following the course, participants were encouraged to use their assigned digital intervention weekly and to engage in standard bag-and-mask skills practice with a manikin at the HBB practice corner set up at their facility. Postcourse assessments were repeated at 1, 3, and 6 months after the class. A follow-up survey was completed.

### Data collection

Data were collected in person by study staff who had completed a HBB second edition master trainer course by experienced HBB master trainers. Staff used the mHBS/DHIS2 tracker app for offline data collection.[26] The mHBS tracker app contained digitised HBB knowledge check, BMV skill check and OSCE A and OSCE B checklist and was used by the participants to report their HBB corner practice. The mHBS trainer app separately tracked educational interventions access and use. To standardise data collection and feedback to study participants, an enhanced neonatal simulator, called *NeoNatalie Live* (Laerdal Medical) was used for BMV. Compared with the low-fidelity *NeoNatalie* simulators used for HBB training (including the HBB practice corners in this study), *NeoNatalie Live* manikin can be programmed to simulate key physiological parameters, such as variable rates of lung stiffness and heart rate and provides auditory and visual cues, in the form of 'crying' and increased heart rate when the end-user provides BMV.[31] In addition, brief automated feedback for the end-user is provided using a Bluetooth-connected tablet device at the end of the assessment as 'well done' or 'needs improvement' based on bag and mask performance. The use of the *NeoNatalie Live* manikin software enabled the correlation of observer collected metrics with manikin collected data. The automated feedback provided by *Neonatalie Live* was the only feedback provided following each assessment.[31]

### Patients or public involvement

Patients or the public were not involved in the design, or conduct, or reporting, or dissemination plans of our research.

### Sample size calculations

We hypothesised that there will be at least a 20% difference in the proportion of subjects who pass OSCE B at the 6-month evaluation between the VR group or video group and the control group. A sample size of 83 subjects per group would provide 80% power to detect a difference in pass rates between groups if the true pass rates were 85% and 65%, respectively, based on a two-sided $\alpha=0.05$. The required total sample size for the three groups (VR, video and control) was 249. We recruited 274 participants total to allow for 10% dropout over the 6-month follow-up period.

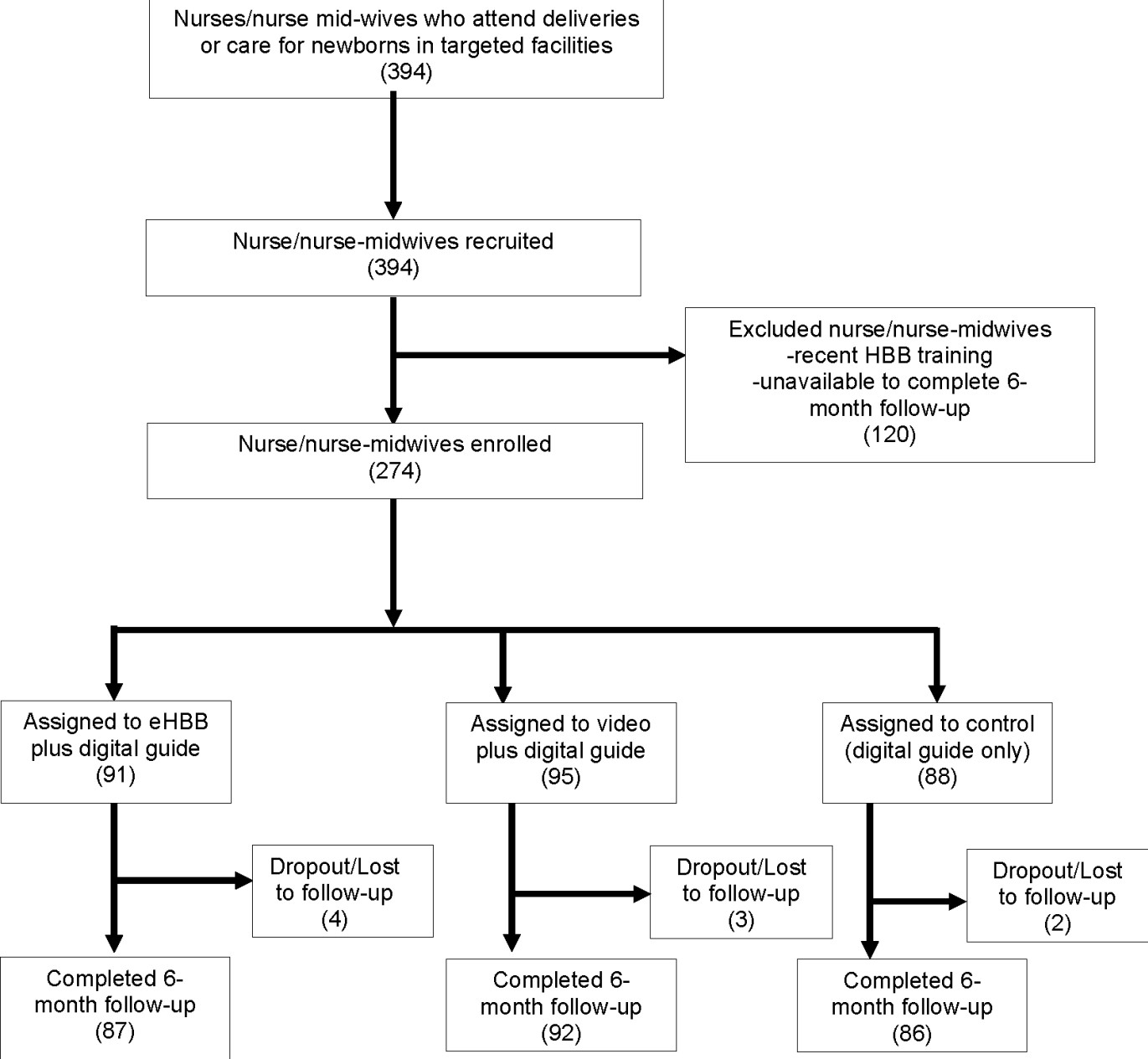

**Figure 2** Consolidated Standards of Reporting Trials diagram. eHBB, electronic Helping Babies Breathe.

## Data analysis

An intention-to-treat analysis was performed, where participants were grouped according to their randomly allocated experimental group (VR, video or control) regardless of their actual exposure. Fisher's exact test was used to test for any differences in pass rates among the three groups for each of the study evaluations: BMV skills assessment, and standardised simulations of routine care and initial resuscitation (OSCE A) and prolonged resuscitation (OSCE B). Post hoc pairwise comparisons and comparisons between demographic groups were also performed using Fisher's exact test. Within-group comparisons of evaluation results between timepoints were performed using the sign test. Participant exposure to the interventions (time in the mHBS trainer app) and

self-reported clinical activity during the follow-up period were compared between experimental groups using the Kruskal-Wallis test and Wilcoxon rank-sum test. All statistical calculations were conducted with the statistical computing language R (V.4.0.0; R Foundation for Statistical Computing, Vienna, Austria). Throughout, two-sided tests were used, with statistical significance defined as $p < 0.05$.

## RESULTS

Recruitment, training and follow-up were conducted concurrently at participating sites from December 2018 to August 2019 with follow-up continuing until February 2020. Of the 394 nurses and nurse-midwives identified

**Table 1**  Demographics of study participants

| Demographics | All n=274 n (%) | VR +digital guide n=91 n (%) | Video+digital guide n=95 n (%) | Digital HBB guide only n=88 n (%) |
|---|---|---|---|---|
| Age, years, mean (SD) | 38 (9) | 37 (9) | 41 (10) | 37 (9) |
| Gender | | | | |
| Female | 250 (91.2) | 82 (90.1) | 88 (92.6) | 80 (90.9) |
| Profession* | | | | |
| Nurse | 133 (48.5) | 43 (47.3) | 46 (48.4) | 44 (50.0) |
| Nurse-midwife | 135 (49.3) | 48 (52.7) | 44 (46.3) | 43 (48.9) |
| Midwife | 5 (1.8) | 0 (0.0) | 5 (5.3) | 0 (0.0) |
| Country | | | | |
| Kenya | 128 (46.7) | 44 (48.4) | 42 (44.2) | 42 (47.7) |
| Nigeria | 146 (53.3) | 47 (51.6) | 53 (55.8) | 46 (52.3) |
| Ward† | | | | |
| Labour/Delivery ward | 188 (71.8) | 62 (71.3) | 63 (70.8) | 63 (73.3) |
| Postnatal ward | 48 (18.3) | 15 (17.2) | 15 (16.9) | 18 (20.9) |
| Newborn unit/NICU | 17 (6.5) | 6 (6.9) | 7 (7.9) | 4 (4.7) |
| Operating theatre | 9 (3.4) | 4 (4.5) | 4 (4.5) | 1 (1.2) |
| Post-training experience (years)‡ | | | | |
| <5 | 65 (23.9) | 22 (24.4) | 17 (18.1) | 26 (29.5) |
| 5–10 | 83 (30.5) | 34 (37.8) | 24 (25.5) | 25 (28.4) |
| 11–15 | 40 (14.7) | 11 (12.2) | 16 (17.0) | 13 (14.8) |
| 16–20 | 32 (11.8) | 11 (12.2) | 13 (13.8) | 8 (9.1) |
| >20 | 52 (19.1) | 12 (13.3) | 24 (25.5) | 16 (18.2) |
| Prior HBB training | 52 (19.0) | 16 (17.6) | 23 (24.2) | 13 (14.8) |
| Owns a smartphone† | 222 (92.5) | 72 (91.2) | 75 (91.5) | 74 (94.9) |

*Missing value=1.
†Missing value=12.
‡Missing value=34.
HBB, Helping Babies Breathe; NICU, neonatal intensive care unit; VR, virtual reality.

who attended deliveries at the participating sites, 274 consented to participate in the study. Of the 274 participants, 265 (97%) completed a 6-month assessment, with a similar dropout rate in each group (p=0.52 for the difference between groups, figure 2).

Most participants were female (91%), nurse-midwives or midwives (51%), who worked in the labour and delivery ward (72%). Nearly all owned a smartphone (table 1). Neonatal resuscitation knowledge and skills assessments were conducted immediately after the in-person course. There were no differences in knowledge check scores (VR 18 (17–18), video 18 (17–18), control 18 (17–18), p 0.76) or pass rates on the BMV (VR 46% (n=83), video 46% (n=84), control 52% (n=79), p=0.72), OSCE A (VR 76% (n=91), 78% (n=95), 72% (n=88), p=0.63) and OSCE B (VR 59% (n=91), video 73% (n=95), control 62% (n=88), p=0.13) assessments across groups on the immediate postcourse assessments.

## Neonatal resuscitation knowledge and skills on follow-up assessments

Neonatal resuscitation skills assessments were conducted at 1, 3 and 6 months after the in-person course. Differences in pass rates on the BMV, OSCE A and OSCE B checklists across groups on the 6-month postcourse assessments were not statistically significant (table 2).

To determine whether pass rates were impacted by years of experience, age, profession, ward or prior HBB training, BMV skills, OSCE A and OSCE B pass rates were compared between groups, one at a time. Participants with <5 years of experience performed better on the OSCE A and OSCE B immediate postcourse assessments (p=0.022 and p=0.034, respectively). Nurse-midwives performed better on BMV skills (p<0.001), OSCE A (p<0.001) and OSCE B immediate postcourse assessments (p=0.011) compared with nurses, although pass rates were similar for all three tests at 6 months (p=0.14–0.89). Ward assignment to newborn unit, neonatal intensive care unit (NICU) or postnatal ward was also associated with greater

**Table 2** Comparison of 6-month postcourse pass rates between groups

| Variable | Group n (%) | | | VR versus control | | | Video versus control | | |
|---|---|---|---|---|---|---|---|---|---|
| | VR (n=87) | Video (n=92) | Control (n=86) | P value† | Δ | (95% CI) | P value* | Δ | (95% CI) | P value* |
| BMV skill check | 24 (28) | 23 (25) | 19 (22) | 0.71 | 5.5% | (−8.5% to 19.5%) | 0.48 | 2.9% | (−10.7% to 16.5%) | 0.72 |
| OSCE A | 66 (76) | 70 (76) | 62 (72) | 0.78 | 3.8% | (−10.5% to 18.0%) | 0.61 | 4.0% | (−10.0% to 18.0%) | 0.61 |
| OSCE B | 54 (62) | 55 (60) | 42 (49) | 0.18 | 13.2% | (−2.6% to 29.1%) | 0.09 | 10.9% | (−4.7% to 26.6%) | 0.18 |

*Fisher's exact test comparing pass rates of the VR or video group with the control group.
†Fisher's exact test comparing pass rates among the three groups.
BMV, bag-and-mask ventilation; OSCE, objective structured clinical examination; VR, virtual reality.

immediate postcourse pass rate on BMV skills (p<0.001), OSCE A (p=0.001) and OSCE B assessments (p<0.001). At the 6-month follow-up assessment, there were no significant differences in BMV skills, OSCE A or OSCE B pass rates by country site, years of experience, age, ward and HBB training >1 year prior to the study (table 3).

### Neonatal resuscitation performance changes over time
There was a decline in performance on neonatal resuscitation skills assessments at 1 month across all groups with a variable degree of recovery of skills by the 3-month and 6-month assessments (figure 3).

### BMV skill pass rates
BMV skills showed a decline at the 1-month assessment and remained significantly lower than the immediate postcourse baseline in all groups at 3 months (−23% VR, p=0.001, −25% video, p<0.001, −31% control, p<0.001) and in the video and control groups at 6 months (−15% VR, p=0.10, −21% video, p<0.01, −27% control, p=0.001).

### OSCE A pass rates
While pass rates decreased on the OSCE A assessments across all groups at 1 month, the groups improved over time and OSCE A pass rates were close to the immediate postcourse baseline at 6 months (−1% VR, −1% video, 0% control, p=0.83), with the VR group demonstrating an earlier recovery of skills (−2% VR, −9% video, −7% control, p=0.52 at 3 months). At 6 months, the VR group showed good performance on questions: *prepares the area for ventilation* and *checks function of bag, mask and suction device* (VR 92% (n=87), video 89% (n=92), control 84% (n=86), p=0.25), *recognises baby is crying and breathing well* (VR 100% (n=87), video 99% (n=92), control 95% (n=86), p=0.07) and *communicates with mother* (VR 94% (n=87), video 86% (n=92), control 86% (n=86), p=0.12), although these differences were not statistically significant.

### OSCE B pass rates
OSCE B pass rates were higher than the immediate postcourse baseline at 3 and 6 months in the VR group (+4% at 3 months, p=0.64; +3% at 6 months, p=0.74) and lower in the video (−21% at 3 months, p<0.001; −14% at 6 months, p=0.07) and control groups (−7% at 3 months p=0.43; −14% at 6 months, p=0.10). Across groups, the performance was sustained on some items of the OSCE B skills checklist that are necessary to improve ventilation such as *reapply mask* and *reposition head*, while other recommended steps such as *clear mouth and nose of secretions, open the mouth and squeeze bag harder*, showed a greater decline in performance (figure 4). On post hoc analysis of OSCE B assessments at the 6-month follow-up, there was a statistically significant difference between the VR and control groups on the frequency of performing the steps: *opens mouth slightly* (54% vs 37%, p=0.03) and *squeezes bag harder* (75% vs 59%, p=0.04) and providing the target *ventilation rate of 30–50 breaths per minute* (86% VR vs 73% control, p=0.04). Differences in performance between the video and control groups were not statistically significant on

**Table 3** Pass rates compared between demographic groups

| Demographic | BMV skills check* | | OSCE A* | | OSCE B* | |
|---|---|---|---|---|---|---|
| | Immediate n (%) | 6-month n (%) | Immediate n (%) | 6-month n (%) | Immediate n (%) | 6-month n (%) |
| **Years of experience** | | | | | | |
| <5 | 36 (62.1) | 18 (28.6) | 57 (87.7) | 49 (77.8) | 50 (76.9) | 37 (58.7) |
| 5–10 | 27 (41.5) | 11 (16.4) | 47 (65.3) | 52 (77.6) | 41 (56.9) | 42 (62.7) |
| 11–20 | 20 (44.4) | 17 (32.7) | 39 (75.0) | 38 (73.1) | 29 (55.8) | 25 (48.1) |
| 20+ | 35 (45.5) | 20 (24.7) | 62 (74.7) | 58 (71.6) | 57 (68.7) | 46 (56.8) |
| P value† | 0.11 | 0.18 | 0.02 | 0.79 | 0.03 | 0.45 |
| **Age (years)** | | | | | | |
| 20–30 | 39 (60.0) | 22 (31.0) | 60 (83.3) | 54 (76.1) | 53 (73.6) | 42 (59.2) |
| 31–40 | 36 (45.0) | 14 (16.7) | 66 (75.9) | 59 (70.2) | 59 (67.8) | 44 (52.4) |
| 41–50 | 29 (42.0) | 20 (26.0) | 54 (66.7) | 61 (79.2) | 50 (61.7) | 49 (63.6) |
| 51+ | 14 (43.8) | 10 (30.3) | 26 (76.5) | 24 (72.7) | 16 (47.1) | 16 (48.5) |
| P value† | 0.16 | 0.16 | 0.13 | 0.60 | 0.05 | 0.36 |
| **Profession** | | | | | | |
| Nurse | 31 (27.7) | 34 (26.6) | 87 (65.4) | 95 (74.2) | 76 (57.1) | 79 (61.7) |
| Nurse-midwife | 87 (64.9) | 32 (23.4) | 119 (84.4) | 103 (75.2) | 102 (72.3) | 72 (52.6) |
| P value† | <0.001 | 0.57 | <0.001 | 0.89 | 0.01 | 0.14 |
| **Ward** | | | | | | |
| Labour/Delivery ward or operating theatre | 77 (42.8) | 49 (24.6) | 147 (71.7) | 150 (75.4) | 120 (58.5) | 113 (56.8) |
| NBU/NICU or postnatal ward | 39 (70.9) | 13 (24.1) | 53 (93.0) | 37 (68.5) | 51 (89.5) | 30 (55.6) |
| P value† | <0.001 | >0.99 | 0.001 | 0.30 | <0.001 | 0.88 |
| **Prior HBB training** | | | | | | |
| Yes | 22 (51.2) | 14 (28.6) | 40 (76.9) | 38 (77.6) | 29 (55.8) | 29 (59.2) |
| No | 96 (47.3) | 52 (24.1) | 166 (74.8) | 160 (74.1) | 149 (67.1) | 122 (56.5) |
| P value† | 0.74 | 0.58 | 0.86 | 0.72 | 0.15 | 0.75 |

*Values are no (%) if not otherwise specified.
†Fisher's exact test comparing pass rates between groups.
BMV, bag-and-mask ventilation; HBB, Helping Babies Breathe; NBU, newborn unit; NICU, neonatal intensive care unit; OSCE, objective structured clinical examination.

these metrics: *opens mouth slightly* (47% vs 37%, p=0.23), *squeezes bag harder* (73% vs 59%, p=0.06) and *ventilation rate of 30–50 breaths per minute* (85% video vs 73% control, p=0.06).

### Participant exposure to interventions and clinical activities
Participants were assigned access to study interventions through the mobile Helping Babies Survive (mHBS) app and were encouraged to self-report HBB corner practice and deliveries assisted during the follow-up period. The median user time spent in the mHBS trainer app was 103 (85–126) min. This software reported metric reflected the time spent accessing the educational interventions in all groups. There was no difference between groups in number of minutes spent using the mHBS trainer 103 (85–126) over the 6-month follow-up period (VR 101 (81–120), video 108 (87–133), control 102 (87–126), p=0.36). There was no significant difference

in self-reported clinical activities with median number of deliveries assisted (VR 25 (5–64), video 25 (12–75), control 28 (7–108), p=0.51) and median deliveries requiring BMV (VR 9 (3–20), video 10 (4–24), control 9 (4–26), p=0.67). The median HBB corner practice days were also similar across groups (VR 16 (6–42), video 20 (7–38), control 16 (7–51), p=0.86)).

### Participant feedback
Participant feedback indicated overall positive impressions of the VR and video refresher training interventions. On a 6-month follow-up survey with a Likert scale of 1–5 with 1 being strongly disagree and 5 being strongly agree, VR group participants agreed/strongly agreed that eHBB VR was easy to use (92%, n=82), realistic (90%, n=81) and provided valuable clinical practice (92%, n=81) and feedback (90%, n=68). A majority of the video group participants also agreed/strongly agreed that the video was easy

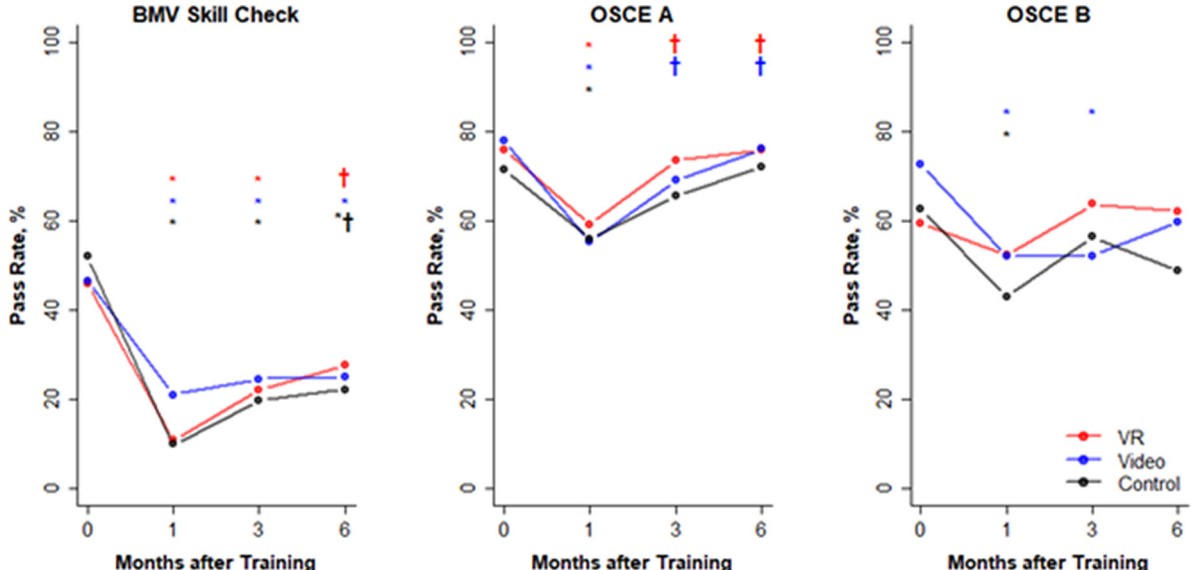

**Figure 3** Pass rates of bag-and-mask ventilation (BMV) skills check, objective structured clinical examination (OSCE) A and OSCE B assessments over time. Pass rates of BMV skills check, OSCE A and OSCE B assessments over time. Immediate postcourse, 1 month, 3 months and 6 months assessments indicated by solid circles. *Statistically significant changes within each experimental group (virtual reality (VR), video and control) from the immediate postcourse assessment. †Significant changes from the 1-month assessment.

to use (88%, n=83), realistic (96%, n=81) and valuable for clinical practice (85%, n=81). If given the opportunity, 95% (n=65) of VR and 98% (n=82) of video respondents would use their assigned intervention again.

## DISCUSSION

This is the first randomised controlled trial that assesses the impact of mobile VR training for neonatal resuscitation skills retention in HCWs in a resource-scarce setting after standard in-person HBB training. Mobile VR training was highly feasible and acceptable to HCWs in a LMIC setting. Previous reports on neonatal resuscitation training using the HBB curriculum have demonstrated a decline in skills within months of training which may interfere with transfer of skills to clinical practice.[14–16 32 33] In this study of digital interventions for neonatal resuscitation skills

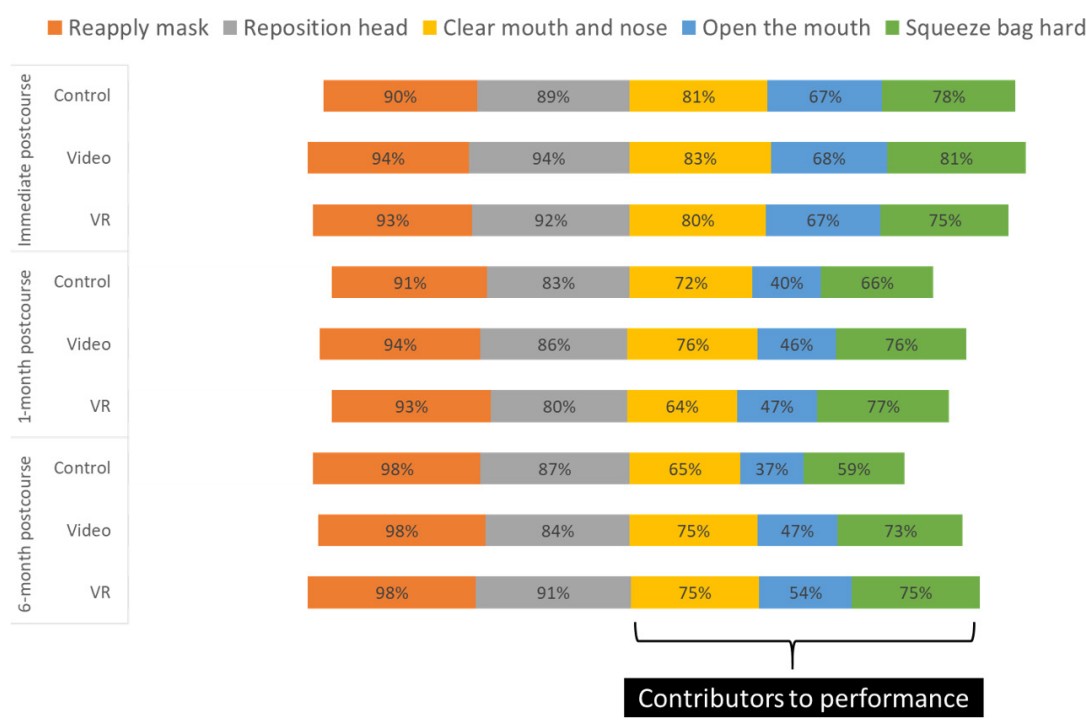

**Figure 4** Objective structured clinical examination (OSCE) B skills performance on critical steps of ventilation. VR, virtual reality.

retention, we found that digital interventions such as VR and video used for refresher training supported the retention of neonatal resuscitation skills in HCWs in Nigeria and Kenya. While the 6-month postcourse performance was similar across groups, when compared with immediate postcourse performance, the decreases in BMV skills pass rates over time were significant in the video and control groups but not in the VR group. Also, contrary to the expected decline in performance over time, OSCE B pass rates were higher at 3 and 6 months than immediately postcourse in the VR group, suggesting that the VR training may provide additional support for the skills needed for prolonged neonatal resuscitation.

Both VR and video have been described for training in HCWs.[24 34–40] Video is a familiar medium but has been long considered a passive learning modality which should be combined with an active learning modality such as manikin-based practice or simulation.[22 41] Virtual simulation, an active learning modality, is thought to support learning through repetition, user engagement and identity formation.[42 43] Although the overall performance of the video and VR groups was similar at the 6-month assessment, the VR group demonstrated an increased performance over time on some neonatal resuscitation skills that have been demonstrated gaps in educational simulation-based settings such as the steps to improve BMV.[44] In settings where in-person refresher training is costly and potentially challenging, and may be even more difficult within the context of COVID-19 concerns, digital and telehealth interventions may adequately support the retention of neonatal resuscitation knowledge and skills.[45]

Little is known about the feasibility and acceptability of VR as a novel educational modality for training HCWs in resource-scarce settings. Previous descriptions of mobile VR use with school-age children in LMICs suggest that mobile VR simulations can be used to demonstrate real-world phenomena, illustrate abstract concepts and motivate learners.[46] After using VR, students asked questions that reflected a deeper level of curiosity, engagement and reflection on lesson topics. They also took ownership of the programme by recharging mobile devices and creating their cardboard viewers.[46] Digital interventions such as VR may provide engaging, individualised and incentivised practice opportunities.[39] A survey of HCWs' perspectives on simulation-based training in Nigeria showed a lack of awareness of VR training, but willingness to use VR simulations if they were available.[47] Computer-based simulations have been used in high-resource settings as an adjunct to in-person neonatal and paediatric resuscitation training.[22] VR may support the transfer of knowledge to practice through interactive learning, problem-solving and standardised feedback.[48 49] Connections that emerge between the participant's offline and in-game identity, and the actual interactions with virtual newborns and mothers within the VR simulation, may modify attitudes and behaviours that relate to clinical practice.[50–53] The HCWs in our study responded positively to mobile VR training.

Recently, Erdsal *et al*[54] described recommendations to improve the implementation of training programmes like HBB by establishing a system for training HCWs and conducting low-dose high frequency practice that is tailored to needs, incentivised and self-reflective. This practice should emphasise both cognitive and psychomotor skills important for successful neonatal resuscitation. For practising healthcare providers, the preparation of delivery and initial steps of resuscitation covered by OSCE A checklist are frequently performed in clinical practice, as approximately 10% of all babies born require some resuscitation.[22] A number of digital innovations have been developed, over the past decade, to support *HBB* education and training.[26 55 56] Based on our findings, after initial training, basic neonatal resuscitation skills may be supported by a range of digital training including VR, video and even digital guide only. However, prolonged resuscitation (represented by the OSCE B scenario) occurs in only 1% of deliveries, so the performance of these skills is less common in clinical practice, particularly in low-volume healthcare facilities. Simulation practice is important for skills retention in HCWs at these facilities and low-dose high frequency practice at a facility-based HBB corner is recommended.[57] Pass rates on bag and mask skills were higher at 6 months in the VR group. Improvements in BMV performance over time were specifically seen in the critical skills needed to improve ventilation in a baby who is not responding. Resuscitation actions to improve ventilation like *opening the mouth* of the baby are notably hard to reinforce on manikin-based training because the manikin's mouth is designed to be always open.[58]

This study had several limitations. We only recruited participants from secondary and tertiary healthcare facilities urban and semi-urban resource-scarce settings. The study findings may not apply to HCWs who work in high-resource settings, at primary healthcare facilities or in rural settings. Although a majority of the study participants reported owning a mobile phone, to ensure uniform access to the study interventions, all participants were provided a study phone to enable access to the digital resources. VR applications are compatible with a wide range of low-cost mobile phones, but not all phones can run VR applications. Participants were asked to access the digital interventions weekly, but the average frequency of access to the application was monthly and may have impacted study findings. The optimal frequency of access is unknown and is an opportunity for future study.

## CONCLUSION

Digital interventions supported the retention of neonatal resuscitation knowledge and skills for HCWs in Nigeria and Kenya. eHBB VR training was highly feasible and acceptable to HCWs in LMICs. eHBB VR may provide additional support for neonatal resuscitation skills retention when compared with other digital interventions.

**Author affiliations**
[1]Department of Pediatrics, University of Washington, Seattle, Washington, USA
[2]Department of Pediatrics, Indiana University School of Medicine, Indianapolis, Indiana, USA
[3]Department of Radiology, University of Washington, Seattle, Washington, USA
[4]Department of Paediatrics, University of Lagos College of Medicine, Lagos, Nigeria
[5]Alupe University College, Busia, Kenya
[6]eHealth4everyone, Abuja, Nigeria
[7]Centre for Tropical Medicine, Nuffield Department of Clinical Medicine, Oxford, UK
[8]Department of BioHealth Informatics, Indiana University-Purdue University at Indianapolis, Indianapolis, Indiana, USA

**Acknowledgements** We would like to acknowledge the senior administrative and nursing staff at healthcare facilities that participated in the study. We acknowledge the input of Dr Susan Niermeyer, Dr Michael Visik and Dr David Bolnick in developing the HBB 2nd edition video. We acknowledge the members of the University of Oxford LIFE Project who assisted in development of the eHBB VR application: Professor Michael English, Dr Niall Winters, Dr Hilary Edgcombe, Mr Jakob Rossner, Ms Naomi Muinga (KEMRI-Wellcome Trust, Kenya) and the staff at National Hospital Abuja who assisted with eHBB VR testing: Dr Amsa Mairami, Dr Fatima Mairami, Dr Adekunle Otuneye, Dr Lamidi Audu and Dr Mariya Mukhtar-Yola. Research assistants were provided with standardised Helping Babies Breathe 2nd Edition Facilitator training by Mr Sammy O. Barasa and Mr Geoffrey Mwai, of the Kenya Helping Babies Survive Master Trainer Corps. We would also like to acknowledge the study team members and collaborators who provided administrative, technical and other support for the study: Ms Oyin Akinrinola, Mr Dillon Afenir, Mr Bhavani Agnikula Kshatriya, Dr Benjamin Al-Haddad, Mr Iman Asangansi, Mr Prem Avanigadda, Mr Ariyo Ayokunle, Dr Brian Bresnahan, Mr Uchechukwu Chinedu, Mr Emeka Chukwu, Mr Chima Chukwudi, Ms Bailey Clopp, Mr Domnan Diretnan, Mr Aniekan Ebito, Ms Sakina Ginary, Ms Chioma Ginikanwa, Mr Oystein Gomo, Mr Alex McGee, Mr Manasseh Mmadu, Ms Ajayi Motunrayo, Mr Alvin Ogbonna, Mr Kevin Otieno, Ms Shruti Patel, Ms Amanda Stiffler, Dr Amy Thompson; in memorium, Dr Charles Spiekerman and Ms Elisabeth Meyers.

**Contributors** RU and SB conceptualised and designed the study, drafted the initial manuscript, designed the data collection tools and reviewed and revised the manuscript. CP and SP designed the data collection tools and reviewed and revised the manuscript. BNE, IBF, FMO, IA, MN, OAO, COA and AM collected data, and reviewed and revised the manuscript. JF and DSH carried out the initial analyses and reviewed and revised the manuscript. CVE and FE conceptualised and designed the study, collected data and critically reviewed the manuscript for important intellectual content. All authors participated in the revision of the manuscript, approved the final version and agreed to be accountable for all aspects of the work.

**Funding** This work was supported by the Bill & Melinda Gates Foundation (grant number OPP1169873). Under the grant conditions of the Foundation, a Creative Commons Attribution 4.0 Generic License has already been assigned to the Author Accepted Manuscript version that might arise from this submission.

**Competing interests** RU and CP developed the eHBB VR application. SB and SP developed the mHBS/DHIS2 application. The other coauthors have no conflicts of interest relevant to this article to disclose.

**Patient consent for publication** Not required.

**Ethics approval** The study was approved by the University of Washington Institutional Review Board (IRB) approval number STUDY00005297, the Indiana University IRB approval number 1807371465, the Moi University, Health Research Ethics Committee approval number 0003109 and the University of Lagos College of Medicine Health Research Ethics Committee approval number CMUL/HREC/09/18/445.

**Provenance and peer review** Not commissioned; externally peer reviewed.

**Data availability statement** Data are available on reasonable request. Deidentified data are available on request from Dr. Rachel Umoren at nestprog@uw.edu.

**ORCID iD**
Rachel Umoren http://orcid.org/0000-0003-2356-9278

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
