## [Reviewer comments · BMJ Open]

ARTICLE DETAILS

TITLE (PROVISIONAL)	eHBB: A randomized controlled trial of virtual reality or video for neonatal resuscitation refresher training in healthcare workers in resource scarce settings
AUTHORS	Umoren, Rachel; Bucher, Sherri; Hippe, Daniel; Ezenwa, Beatrice; Fajolu, Ireliola; Okwako, Felicitas; Feltner, John; Nafula, Mary; Musale, Annet; Olawuyi, Olubukola; Adeboboye, Christianah; Asangansi, Ime; Paton, Chris; Purkayastha, Saptarshi; Ezeaka, Chinyere; Esamai, Fabian

VERSION 1 – REVIEW

REVIEWER	Wood, Jessica The University of Newcastle, School of Nursing and Midwifery
REVIEW RETURNED	26-Feb-2021

GENERAL COMMENTS	Minor Revisions regarding article titled eHBB: A randomized controlled trial of virtual reality or video for neonatal resuscitation refresher training in healthcare workers by Umoren et al.	
	Page and line number	Feedback to be addressed
	P.1, line 1-2	Title: this study was conducted in low-resource/resource scarce areas. I feel it is important this is identified in the title of the article
	P.1, line 28	Keywords: consider adding 'virtual reality' as a keyword
	P.2, line 47	Expand HBB abbreviation since this is the first time it has been mentioned
	P.2, line 47-48	Abstract objective: add "in low resource settings" or "in resource scarce settings" to the end of the Objective statement
	P.2, line 61	Needs to say 60% (only says 60)
	P.2, line 81-83	It needs to be acknowledged that the study findings also may not apply in high resource countries/settings
	P.3, line 96	Please clarify what is meant by 'traditional methods'. What does this include?
	P.3, line 107	Expand HCW abbreviation since this is the first time it has been mentioned
	P.5, line 136	There is an accidental full stop placed between the words 'course by'
	P.5, line 139	Unclear what DHIS2 stands for – please expand abbreviation or explain what DHIS2 app is
	N/A	Comment in relation to Methods section: it would be valuable to have more information about the VR intervention as it would be difficult for this

		study to be replicated without having a clearer idea of what is involved with the use of this app. What is involved in the VR simulation? Is there one repeating scenario that HCW's practice? Are the images in 3D? How interactive is it? Does it require an internet connection to use? Screenshots from inside the app would be helpful.
	P.6, line 173	Should say heart rate, not just heart
	P.7, line 198	The self-reported requirement for clinical activity and practice at the HBB practice corner needs to be acknowledged as a limitation as there is the potential for reporting errors
	P.8, line 208	Missing bracket at the end of (97%)
	P.8, line 214	This statement indicates 51% were nurse-midwives, but Table 1 shows this as being 49.3% - please correct
	P.9 (Table 1)	There are some columns here that do not add up to 100%, for example, the Profession section under the Digital HBB Guide only column totals to 98.9%. It is not indicated in this section that there are any missing values. Please check this table.
	P.9 (Table 1)	The Legend under the table has NBU written, but this abbreviation is not used in the table. This can be removed.
	P.10, line 243	Now NBU is mentioned in text so this needs to be expanded first before being abbreviated
	P.11, Table 3	Should the OSCE B column also have an * on it since you have indicated percentages in this column?
	P.13, line 291	You have provided the metrics for the VR and control group and it would be good to show the Video and control group metrics despite not being statistically significant
	P.16, line 366	Where is this statistic from? 10% of babies in Africa? Globally? Please specify and cite source after this sentence (could not clearly identify in the reference list which source this came from)
	P.16, line 373	You state that simulation practice is important for skills retention in HCWs – perhaps you could add to this that access to 'corner practice' at facilities could be more widely implemented. Was this regularly accessed by the participants in the research?
	p.17, lines 378-386	Limitations: It needs to be acknowledged here that the results may not be generalisable to high-resource settings. It is acknowledged in lines 384-385 that digital interventions were accessed monthly. No VR or video analytics have been reported in this paper. How was analytics and frequency of use recorded/tracked? Was it via software or self-reporting? The minimal frequency of use of the digital interventions is a limitation that should be expanded on more
	p.19-22	Reference list: Please be consistent with referencing, particularly with journal sources. Some journal names have all words

		capitalised (e.g. reference no. 23) while others only have the first word in the name capitalised (e.g. reference no. 21). Some journal names are abbreviated (e.g. reference no. 3 and no. 8) while others are written in full. Please adjust so all sources are consistent.
	General comment	It would have been good to see what is in the OSCE A and OSCE B checklists

REVIEWER	Barré, Jessy Universite de Paris
REVIEW RETURNED	01-Mar-2021

GENERAL COMMENTS	Thank you for this paper. The study is very interesting (neonatal resuscitation skills in Nigeria and Kenya healthcare workers) and current (virtual reality training). I have a few remarks which you will find below. Please explain more the VR material/system. A photo is welcomed (if authorized). Page 3 and 4: explain NR acronym. “Participants were encouraged to use their assigned digital intervention weekly and to engage in standard bag-and-mask skills practice with a manikin at the HBB practice corner set up at their facility” (I159): Did you check the auto-training (deliberate practice) of each participant between evaluations (1/3 and 6 months)? that could be created differences between groups. “Standardized knowledge and skills assessments were conducted by trained research assistants?” (I148) : Who are they? How much? Did you realized an inter-rater evaluation? Is there statistical differences between demographics information in Table 1? (e.g. differences between nurses and midwives scores, etc.). Also, please explain more the scoring system (table 2 and course assessment part): maximum score/rating, etc. Table 3: it is “n (%)” in the table? please indicate. How did you explain the decrease of skills between first and second session? In the same way, why there are no differences between groups in the BMV and OSCE A scores? what do you think? The training module differs by modalities (VR, video), but may be also by content? More explanations/infomration of these modules are welcomed. At last, why this VR training are not available for all country? Some virtual environments (e.g : VR, screen based simulation), are designed for all (see Michelet, D., Barre, J., Truchot, J., Piot, M. A., Cabon, P., & Tesniere, A. (2020). Effect of Computer
---

	Debriefing on Acquisition and Retention of Learning After Screen-Based Simulation of Neonatal Resuscitation: Randomized Controlled Trial. JMIR Serious Games, 8(3), e18633.). May be there is some limits in the software? what are they? etc. Thanks.
--	---

VERSION 1 – AUTHOR RESPONSE

Response to Reviewers Comments

Reviewer 1

Dr. Jessica Wood, The University of Newcastle

Comments to the Author:

Thank you for the opportunity to review this article. The article is well presented, clear in its objectives and focus, and has a sound discussion of the outcomes and their relevance. There are some minor amendments to attend to, mainly grammatical issues which I have attached as a document below.

Thank you, Dr. Wood, for your thoughtful review. We have included in the table below, our responses to your comments.

Page and line number	Feedback to be addressed	Author Response
P.1, line 1-2	Title: this study was conducted in low-resource/resource scarce areas. I feel it is important this is identified in the title of the article	We have updated the title to reflect this.
P.1, line 28	Keywords: consider adding 'virtual reality' as a keyword	We have included virtual reality as a key word
P.2, line 47	Expand HBB abbreviation since this is the first time it has been mentioned	We have expanded the abbreviation "HBB" to read Helping Babies Breathe on first use.
P.2, line 47-48	Abstract objective: add "in low resource settings" or "in resource scarce settings" to the end of the Objective statement	This has been done.

P.2, line 61	Needs to say 60% (only says 60)	This has been corrected.
P.2, line 81-83	It needs to be acknowledged that the study findings also may not apply in high resource countries/settings	We have included this as a limitation under the Strengths and Limitations of this study.
P.3, line 96	Please clarify what is meant by 'traditional methods'. What does this include?	We have clarified "traditional methods" to read: "the traditional approach of small group facilitated classroom training"
P.3, line 107	Expand HCW abbreviation since this is the first time it has been mentioned	We have expanded the abbreviation "HCW" to read Healthcare workers on first use.
P.5, line 136	There is an accidental full stop placed between the words 'course by'	This has been corrected.
P.5, line 139	Unclear what DHIS2 stands for – please expand abbreviation or explain what DHIS2 app i	We have expanded the abbreviation "DHIS2" to read "District Health Information System"

		on first use.
N/A	Comment in relation to Methods section: it would be valuable to have more information about the VR intervention as it would be difficult for this study to be replicated without having a clearer idea of what is involved with the use of this app. What is involved in the VR simulation? Is there one repeating scenario that HCW's practice? Are the images in 3D? How interactive is it? Does it require an internet connection to use? Screenshots from inside the app would be helpful.	Additional description of the eHBB application has been included in the Methods section including number of scenarios, type of images, interactivity and internet connectivity requirements. Due to space constraints, we have referenced a paper with a detailed description and screenshots of the eHBB application.
P.6, line 173	Should say heart rate, not just heart	This has been corrected.
P.7, line 198	The self-reported requirement for clinical activity and practice at the HBB practice corner needs to be acknowledged as a limitation as there is the potential for reporting errors	This has been included in the limitations.

P.8, line 208	Missing bracket at the end of (97%)	This has been corrected.
P.8, line 214	This statement indicates 51% were nurse-midwives, but Table 1 shows this as being 49.3% - please correct	This has been corrected.
P.9 (Table 1)	There are some columns here that do not add up to 100%, for example, the Profession section under the Digital HBB Guide only column totals to 98.9%. It is not indicated in this section that there are any missing values. Please check this table.	We have updated Table 1 to indicate 1 missing value under the Profession section.
P.9 (Table 1)	The Legend under the table has NBU written, but this abbreviation is not used in the table. This can be removed.	We have removed NBU from the legend.
P.10, line 243	Now NBU is mentioned in text so this needs to be expanded first before being abbreviated	This has been corrected.
P.11, Table 3	Should the OSCE B column also have an * on it since you have indicated percentages in this column?	We have included an * on the OSCE B column.
P.13, line 291	You have provided the metrics for the VR and control group and it would be good to show the Video and control group metrics despite not being statistically significant	The metrics for the Video and control group metrics have been included.
P.16, line 366	Where is this statistic from? 10% of babies in Africa? Globally? Please specify and cite source after this sentence (could not clearly identify in the reference list which source this came from)	The source of this statistic has been cited.
P.16, line 373	You state that simulation practice is important for skills retention in HCWs – perhaps you could add to this that access to ‘corner practice’ at facilities could	This has been added as a recommendation. The self-report

		data on HBB corner practice was reported in the Participant exposure to interventions and clinical activities section. The median HBB corner practice days were 16-20 days over the 6 month study period.
--	--	--

	be more widely implemented. Was this regularly accessed by the participants in the research?	
p.17, lines 378-386	Limitations: It needs to be acknowledged here that the results may not be generalisable to high-resource settings. It is acknowledged in lines 384-385 that digital interventions were accessed monthly. No VR or video analytics have been reported in this paper. How was analytics and frequency of use recorded/tracked? Was it via software or self-reporting? The minimal frequency of use of the digital interventions is a limitation that should be expanded on more	This limitation has been acknowledged. The mHBS Trainer app was utilized by all participants to access the study interventions. The overall user time spent in the mHBS trainer app 103 (85-126) minutes and did not differ between groups. This is a software reported metric. We have added a clarifying

		sentence in the Participant exposure to interventions and clinical activities section. A statement indicating that the frequency of use of the digital interventions may have impacted study findings is included in the limitation section. The optimal frequency of access to educational interventions is unknown.
p.19-22	Reference list: Please be consistent with referencing, particularly with journal sources. Some journal names have all words capitalised (e.g. reference no. 23) while others only have the first word in the name capitalised (e.g. reference no. 21). Some journal names are abbreviated (e.g. reference no. 3 and no. 8) while others are written in full. Please adjust so all sources are consistent.	The reference list has been updated.
General comment	It would have been good to see what is in the OSCE A and OSCE B checklists	The OSCE A and OSCE B checklists are available online and have been referenced.

Reviewer: 2

Dr. Jessy Barré, Université Paris Descartes

Comments to the Author:

Thank you for this paper. The study is very interesting (neonatal resuscitation skills in Nigeria and Kenya healthcare workers) and current (virtual reality training). I have a few remarks which you will find below.

Please explain more the VR material/system. A photo is welcomed (if authorized).

Additional description of the eHBB application has been included in the Methods section including number of scenarios, type of images, interactivity and internet connectivity requirements. Due to space constraints, we have referenced a paper with a detailed description and screenshots of the eHBB application.

Bucher SL, Cardellichio P, Muinga N, et al. Digital health innovations, tools, and resources to support Helping Babies Survive programs. *Pediatrics* 2020;146(Supplement 2):S165-S82.

Page 3 and 4: explain NR acronym.

The NR acronym has been replaced by “neonatal resuscitation” throughout.

“Participants were encouraged to use their assigned digital intervention weekly and to engage in standard bag-and-mask skills practice with a manikin at the HBB practice corner set up at their facility” (I159): Did you check the auto-training (deliberate practice) of each participant between evaluations (1/3 and 6 months)? that could be created differences between groups.

Participants self-reported their practice at the HBB corner on the mHBS Tracker app. Software data tracking on the mHBS Trainer app captured the time spent on the educational intervention. There were no differences between the three study groups on either of these metrics. This data is reported in the Participant exposure to interventions and clinical activities section.

“Standardized knowledge and skills assessments were conducted by trained research assistants?” (I148) : Who are they? How much? Did you realized an inter-rater evaluation?

Study staff had completed a HBB 2nd edition master trainer course by experienced HBB master trainers. The use of the NeoNatalie Live manikin software enabled the correlation of observer collected metrics with manikin collected data. For example:

	BMV checklist Head Tilt Item		
	Checked	Not checked	P
Manikin data			
Median % ventilations with a head tilt	98%	60%	<0.001

Is there statistical differences between demographics information in Table 1? (e.g. differences between nurses and midwives scores, etc.).

The statistically significant differences between demographics (including profession) and performance at different time points in the study are shown in Table 3.

Also, please explain more the scoring system (table 2 and course assessment part): maximum score/rating, etc.

The pass rates were calculated based on previously published tools developed by the American Academy of Pediatrics for the Helping Babies Breathe second edition curriculum.

Table 3: it is “n (%)” in the table? please indicate.

We have included “n (%)” in Table 3.

How did you explain the decrease of skills between first and second session?

The decrease of skills from the immediate post-training assessment to the 1 month assessment was expected and consistent with previous literature. We have added a reference to a systematic literature review on short-term neonatal resuscitation skills in HCW following training, 8 of 10 studies showed significant performance falloff or varying levels of retention within weeks to months of training.

Reisman J, Arlington L, Jensen L, Louis H, Suarez-Rebling D, Nelson BD. Newborn Resuscitation Training in Resource-Limited Settings: A Systematic Literature Review. *Pediatrics*. 2016;138(2):e20154490.

Similar findings have been described in other reports also included in the references.

Arlington L, Kairuki AK, Isangula KG, et al. Implementation of "Helping Babies Breathe": A 3-Year Experience in Tanzania. *Pediatrics*. 2017;139(5).

Eblovi D, Kelly P, Afua G, Agyapong S, Dante S, Pellerite M. Retention and use of newborn resuscitation skills following a series of helping babies breathe trainings for midwives in rural Ghana. *Glob Health Action*. 2017;10(1):1387985.

Goudar SS, Somannavar MS, Clark R, et al. Stillbirth and newborn mortality in India after helping babies breathe training. *Pediatrics*. 2013;131(2):e344-352.

Musafili A, Essen B, Baribwira C, Rukundo A, Persson LA. Evaluating Helping Babies Breathe: training for healthcare workers at hospitals in Rwanda. *Acta Paediatr*. 2013;102(1):e34-38.

In the same way, why there are no differences between groups in the BMV and OSCE A scores? what do you think?

The sample size calculation was based on the OSCE B, a more cognitively challenging assessment of care of the newborn who requires prolonged resuscitation. The BMV and OSCE A scores represent different skills (bag and mask ventilation only and basic resuscitation – stimulation and opening the airway). It is possible that a larger sample size is needed to capture differences between groups in these areas.

The training module differs by modalities (VR, video), but may be also by content? More explanations/information of these modules are welcomed.

The content of the intervention was designed and developed by the study team with input from national experts with the neonatal resuscitation video containing similar content as the VR application. Additional information on the content and links are provided.

At last, why this VR training are not available for all country? Some virtual environments (e.g : VR, screen based simulation), are designed for all (see Michelet, D., Barre, J., Truchot, J., Piot, M. A., Cabon, P., & Tesniere, A. (2020). Effect of Computer Debriefing on Acquisition and Retention of Learning After Screen-Based Simulation of Neonatal Resuscitation: Randomized Controlled Trial. *JMIR Serious Games*, 8(3), e18633.). May be there is some limits in the software? what are they? etc.

We agree that some software applications may have limited availability. The eHBB VR application is available for free download through the Google playstore to learners anywhere in the world. We have included this link in the references.

VERSION 2 – REVIEW

REVIEWER	Wood, Jessica The University of Newcastle, School of Nursing and Midwifery
REVIEW RETURNED	28-May-2021

GENERAL COMMENTS

Thank you to the authors for making all required revisions to this article as requested. I am now satisfied with the information presented in this article and happy to provide an 'Accept' recommendation.